# Antioxidants and Phenolic Acid Composition of Wholemeal and Refined-Flour, and Related Biscuits in Old and Modern Cultivars Belonging to Three Cereal Species

**DOI:** 10.3390/foods12132551

**Published:** 2023-06-29

**Authors:** Grazia Maria Borrelli, Valeria Menga, Valentina Giovanniello, Donatella Bianca Maria Ficco

**Affiliations:** Consiglio per la Ricerca in Agricoltura e l’Analisi dell’Economia Agraria—Centro di Ricerca Cerealicoltura e Colture Industriali, 71122 Foggia, Italy; graziamaria.borrelli@crea.gov.it (G.M.B.); valeria.menga@crea.gov.it (V.M.); v.giovanniello@tiscali.it (V.G.)

**Keywords:** cereals, wholemeal, biscuits, phenolics, phenolic acid compositions, carotenoids, consumer acceptance

## Abstract

Cereals are a good source of phenolics and carotenoids with beneficial effects on human health. In this study, a 2-year evaluation was undertaken on grain, wholemeal and refined-flour of two cultivars, one old and one modern, belonging to three cereal species. Wholemeal of selected cultivars for each species was used for biscuit making. In the grain, some yield-related traits and proteins (PC) were evaluated. In the flours and biscuits, total polyphenols (TPC), flavonoids (TFC), proanthocyanidins (TPAC), carotenoids (TYPC) and antioxidant activities (DPPH and TEAC) were spectrophotometrically determined, whereas HPLC was used for the composition of soluble free and conjugated, and insoluble bound phenolic acids. Species (S), genotype (G) and ‘SxG’ were highly significant for yield-related and all antioxidant traits, whereas cropping year (Y) significantly affected yield-related traits, PC, TPC, TPAC, TEAC and ‘SxGxY’ interaction was significant for yield-related traits, TPAC, TYPC, TEAC, DPPH and all phenolic acid fractions. Apart from the TYPC that prevailed in durum wheat together with yield-related traits, barley was found to have significantly higher values for all the other parameters. Generally, the modern cultivars are richest in antioxidant compounds. The free and conjugated fractions were more representative in emmer, while the bound fraction was prevalent in barley and durum wheat. Insoluble bound phenolic acids represented 86.0% of the total, and ferulic acid was the most abundant in all species. A consistent loss of antioxidants was observed in all refined flours. The experimental biscuits were highest in phytochemicals than commercial control. Although barley biscuits were nutritionally superior, their lower consumer acceptance could limit their diffusion. New insights are required to find optimal formulations for better nutritional, sensorial and health biscuits.

## 1. Introduction

Phenolic compounds and carotenoid pigments, specialized metabolites synthesized during plant development and in response to stress conditions [1], are excellent oxygen radical scavengers. Their intake through the whole-cereal products offers potential health benefits in many chronic diseases [2].

Cereals are a good source of phenolic compounds and carotenoid pigments and, being important components of the human diet, they can contribute to a significant supply of these molecules [3,4].

Phenolic compounds are mostly concentrated in the outer layers of the grain, mainly pericarp and aleurone, and germ [5,6,7]. Adom et al. [8] showed that the bran/germ fraction of wheat contributes 83% of the total phenolic content of the wholemeal flour. The total phenolic content of bran/germ fractions is 15- to 18-fold higher than that of respective endosperm fractions that contribute only 17% of the total phenolic content. Since external layers are lost during roller-milling, the phenolic compounds are scarce in refined cereal products, and only by consuming wholemeal flour it is possible to benefit from their full levels. Moreover, phenolic compounds may also have an impact on color, flavor and astringency, becoming crucial for the acceptability of the final products by the consumer [9]. Among the phenolic compounds, phenolic acids are the most representative in cereals [10].

As previously observed by other authors [10,11,12], phenolic acids may mainly occur as insoluble bound linked to cell-wall constituents and as soluble conjugated forms esterified to sugars and to other low molecular weight components. Only 0.5–2% of phenolic acids exist as soluble free forms. Their structural diversity influences the bioavailability: free phenolic acids can cross the intestinal barrier and be found in the blood, while the bound forms of phenolic acids are scarcely digested and recovered in the feces, and only a small part can reach the colon where it exerts its antioxidant activity [13,14]. Unlike phenolics, carotenoids are one of the most important pigments occurring in nature. Several health benefits have been attributed to carotenoids, including the role as provitamin A and antioxidant activity [15]. In cereals, carotenoids are differently distributed in the kernel: α- and β-carotene are mainly located in the germ while lutein, the most abundant pigment, is equally distributed across the kernel [6,7,16,17,18]. In durum wheat, they are an important quality trait for industry of semolina/couscous and end-products [19,20].

The amount of these specialized metabolites in cereals is highly variable and mostly related to species and variety [8,10,21,22,23]. Žilić et al. [22], analyzing the antioxidant content over one year in a cereal collection, showed that total phenolic and total flavonoid content was higher in hull-less barley, followed by hull-less oat, rye, durum wheat and bread wheat. Interestingly, the highest antioxidant activity observed in hull-less barley was ascribed to a specific subclass of flavonoids, being more effective as antioxidants than vitamin C, E and carotenoids [24,25]. Comparing old and modern durum wheat genotypes, some authors suggested that breeding has qualitatively influenced the profiles of phenolic compounds [26]. Little information is available on this matter in old and modern genotypes, characterized by different year of release and yield potential, in multi-cereal species for more crop years.

Furthermore, environmental factors (E, including year, location, as well as agronomic practices), genetic (G) effects and ‘GxE’ all contribute to determining phenotypic variation for phenolics, with environmental effects larger than genotypic differences [10,27,28]. Contrarily, for carotenoid pigments a strong G effect was evidenced, particularly in durum wheat [29].

Among cereal species, durum wheat, emmer wheat and barley are an important source of carbohydrates for human consumption. Durum wheat is the preferred raw material for pasta making, couscous and some types of bread, mainly cultivated in Southern regions of Italy. According to the year of release, durum wheat cultivars were grouped into modern developed after the introduction in dwarfing genes in the 1950s, and into old those developed before that time [30]. Old cultivars are characterized by greater rusticity and lower yield while the modern ones differ in term of better yield and quality [31]. Previous research findings showed that the total polyphenol content in both old and modern durum wheat cultivars was similar, but the old cultivars had a higher number of unique compounds not observed in modern varieties [32].

In the last decades, farmers and consumers addressed much attention to emmer wheat, which is phylogenetically related to durum wheat [33]. This renewed interest for emmer is mostly due to the grain bioactive substances and to the possibility of using conventional or organic farming practices with low chemical inputs [34]. Especially, the old cultivars, although lower yielding than modern ones, are suitable for the development of more sustainable crop systems [34].

Barley has been recognized for its adaptability to both highly productive agricultural systems and marginal area. It is also high in dietary fiber (mainly β-glucans), minerals and other phytochemicals such as phenolic acids and flavonoids [24,35]. In particular, proanthocyanidins, the major types of flavonoids in barley grain, are oligomeric and polymeric flavan-3-ols that exert strong antioxidant activity and known also for their ability to bind proteins affecting sensory acceptability [36]. Barley cultivars may have yellow, blue or purple color caused by accumulation of flavonoids compounds in distinct layers of grain [37].

Naked (hull-less) barley is a form of domesticated barley that have an easier-to-remove hull, thanks to which it could have multiple food applications for human consumption in bread preparation, breakfast snacks and beverages (alcoholic and nonalcoholic) [38].

The aim of this research is: To explore the differences, in a 2-year evaluation, of phytochemicals as phenolics and carotenoids, and antioxidant activity in wholemeal and refined-flour, of old and modern cultivars belonging to three cereal species, durum wheat, emmer wheat and barley; to evaluate the effect of species, genotype, environment and their interaction on these traits; to study the antioxidants in the biscuits obtained by selected cultivars for each species; and to provide information on consumer acceptability of monovarietal biscuits in comparison with a commercial product.

## 2. Materials and Methods

### 2.1. Plant Materials

Three species, durum wheat (*Triticum turgidum*, ssp. *durum* Desf.), hulled emmer wheat (*Triticum turgidum* L. spp. *dicoccum Shrank*) and barley (*Hordeum vulgare* L. spp. *vulgare*), available at Research Centre for Cereal and Industrial Crops (CREA-CI), were cropped in Foggia (southern Italy) at the experimental fields of the CREA-CI (41°28′ N, 15°34′ E; 76 m a.s.l.), over two crop years (2015–2016 and 2016–2017). For each species, old and modern cultivars were chosen (Table 1). The seeds were planted in 10 m^2^ plots according to a Randomized Complete Block Design with three replications. Standard cultural practices for each species were applied. Meteorological data on two crop years were obtained from an on-site weather station (Appendix A). The plants were harvested mechanically after physiological maturity. All seeds were stored at 4 °C until further processing.

### 2.2. Wholegrain Analysis

Protein content (PC) corrected for dry matter (DM) and Test weight (TW) were determined by NIR (Infratec Nova Analyzer, Foss Italia, Padova, Italy). Thousand Kernel Weight (TKW) was calculated from the mean weight of three sets of 100 grains per plot for each sample.

### 2.3. Processing

#### 2.3.1. Flours Production

To produce the wholemeal flour, the kernels of each cereal species were ground in a sample mill (Tecator Cyclotec 1093; Foss Italia, Padova, Italy) using a 0.5 mm sieves. To obtain semolina in durum wheat or refined flours in other cereals, the seeds were conditioned at 16.5% (wet basis) moisture and were milled at experimental mill (Labormill 4RB, Bona, Monza, Italy) with four rolls and 42 and 54 GG sieves (sieve 180 μm), which separates flour from bran and germ. Henceforth, the semolina and the other flours were named refined flours.

#### 2.3.2. Biscuit Production

Fortore (durum wheat), Molisano (emmer) and L94 (barley) were selected for interesting levels of phytochemicals and used for biscuit production. Grains were milled to wholemeal flour by means of a granite stone mill (diameter 300 mm model Getreidemühle, Colombini Sergio s.a.s, Abbiategrasso, Milan, Italy). Commercial control (CTRL, 100% wheat) was used in the experimentation. Sucrose, eggs, sunflower oil, salt and vanilla essence were purchased at local retailers. Three independent biscuit-making production trials were performed by Frasca Bakery (Foggia, Italy), involved in the present experiment. The biscuit-making process consisted of: (i) kneading for 3 min sucrose (400 g), sunflower oil (320 mL), eggs (4), salt (4 g), vanilla essence and baking (20 g) by an electric mixer with flat beater (PL16 5B, Conti s.r.l, Bussolengo (Verona, Italy), then adding 1 kg wholemeal flour and kneading for 3 min, and finally adding water (250 mL) and kneading for 3 min; (ii) the dough was rolled out on a tray using a rolling pin and cut into desired shapes using a biscuit cutter; (iii) baking in a steam tube deck oven (Mondial 43, Mondial Forni spa, Verona, Italy) for 15 min at 180 °C. Biscuits were finely crushed in a mortar for subsequent analyses.

### 2.4. Chemical Compounds

#### 2.4.1. Carotenoids

Total carotenoids pigments, referred to as yellow pigments (TYPC), were analyzed according to method 14–50 of AACC International, as modified by in Beleggia et al. [39] for microsamples. The data were expressed as micrograms per gram on dry matter (µg g^−1^ DM). All assays were conducted in triplicate.

#### 2.4.2. Phenolics

Phenolic compounds were extracted according to Suriano et al. [40], with minor modifications. The samples (0.5 g) were extracted using 10 mL methanol (80:20) acidified with 1% 12 N HCl, for 30 min in an ultrasonic bath. After centrifugation, the supernatants were used for the determination of phenolics and antioxidant activity. Total polyphenol content (TPC) was determined using Folin–Ciocalteu reagent, according to the modified method of Suriano et al. [40], and expressed as µg gallic acid equivalents (GAE) g^−1^ DM. Total flavonoid content (TFC) was determined according to the method of Kim et al. [41], and expressed as µg catechin equivalents (CE) g^−1^ DM. The total proanthocyanidins (TPAC) were determined according to the modified vanillin assay of Sun et al. [42], and expressed as µg catechin equivalents (CE) g^−1^ DM. All assays were conducted in triplicate.

#### 2.4.3. Phenolic Acid and Flavonoid Composition

Soluble free and conjugated, and insoluble bound phenolic acids and flavonoids were extracted, separated and quantified according to the method described in Suriano et al. [40], with some modifications, using an Agilent 1200 Series HPLC system (Agilent Technologies, Waldbronn, Germany) equipped with a diode array detector. Separation of phenolic acids was achieved using a reversed phase C18 column (InfinityLAB Poroshell 120 RC-C18, 100 × 2.1 mm; particle size = 2.7 μm) from Agilent (Santa Clara, CA, USA). The column temperature was 35 °C, and the mobile phase consisted of (A) water with phosphoric acid 10^−3^ M and (B) acetonitrile at a flow rate of 0.5 mL/min, using the following linear gradient program: 5% B for 2 min, from 5% to 30% B for 10 min, from 30% B to 55% B for 1 min, from 55% to 70% for 2 min, isocratic at 70% for 1 min, linear gradient from 70% to 5% B for 6 min. Two microliters of sample were injected using an autosampler. The wavelengths used for quantification of the phenolic acids were 280 and 320 nm. The quantification was based on the peak area of the following standards: p-Hydroxybenzoic acid, Vanillic acid, Caffeic acid, Syringic acid, Vanillin, Ferulic acid, Sinapic acid, p-Coumaric acid, Protocatechuic acid, Trans-cinnamic acid and Cis-cinnamic acid and Syringaldeide. Moreover, some standards of flavonoids in cereals were used: Quercitin, Catechin and Naringenin. An example of phenolic and flavonoid chromatograms during the whole cereal food supply chain was reported in Appendix A. All used reagents were obtained from Merk Life Science S.r.l, Milano, Italy. All assays were made in triplicate.

#### 2.4.4. Antioxidant Activities

The antioxidant activity was determined using two different assays: the DPPH and TEAC methods. DPPH radical scavenging capacity was determined according to Suriano et al. [40], using a Trolox calibration curve, and measuring the absorbance at 517 nm. Data were expressed as µmol Trolox equivalents (TE) g^−1^ DM. The TEAC Trolox equivalent antioxidant capacity was determined according to the method of Fares et al. [43], by using a Trolox standard curve, on the basis of the percentage inhibition of absorbance at 734 nm of the radical cation ABTS^•+^ and expressed as µmol Trolox equivalents (TE). All assays were conducted in triplicate.

### 2.5. Consumer Acceptance

A sensory evaluation questionnaire was used in this study to assess the degree of liking of the different biscuits based on their sensory appeal, with respect to the CTRL. Thirty untrained participants performed the test, aged between 25 and 65 years, 80% females and 20% males. The sensory attributes evaluated were odor, sweetness, flavor, crumbliness, crispness, color and could also include overall acceptability. A five-point hedonic scale was used to evaluate the attributes for consumer acceptance, varying from disliked extremely (1) to liked extremely (5) [44]. Biscuits were coded with 4 random letters and water was served to participants for mouth cleaning between samples evaluation.

### 2.6. Statistical Analysis

For all the datasets, one-way analysis of variance (ANOVA) was performed to estimate differences ascribable to the species (S), genotype (G) or year (Y) effect, while two-factor ANOVA was applied to study the effect of the ‘SxG’ and ‘SxGxY’ interactions. Whenever a significant F value was obtained for single factors or their interaction, Tukey HSD test was performed at *p* < 0.05 level. Pearson correlations (r) of the means among phenolics and antioxidant activities were calculated. Statistical analyses were performed using the STATISTICA program (StatSoft Italia srl, vers. 8.0, 2007). A Principal Component Analysis (PCA) was performed using a correlation matrix to visualize differences and similarities of PC, yield-related traits and antioxidants in the three species for two years by using the JMP software (SAS Institute Inc., Cary, NC, USA version 8).

## 3. Results and Discussion

### 3.1. Whole Grain Quality and Yield-Related Traits

The characterization of the grains of all samples, with regard to PC and yield-related parameters, TW and TKW, was performed and the results were shown in Table 2.

ANOVA showed significant interactions between species and genotype ‘SxG’ for PC and yield-related traits. Considering the PC, a pronounced effect due to the species (S) was found with a minor significant effect of the year (Y) (*p* < 0.05). Among durum wheats, the old cultivar was characterized by higher protein percentage compared with modern one, as a consequence of breeding programs for higher yields at the expense of grain quality [31,45,46]. The yield increase, essentially due to a greater carbon availability to the grains, is accompanied by the decrease in protein content, by dilution effects [47]. The opposite was observed in emmer wheat and barley. According to Geisslitz et al. [48], higher proteins were observed for ancient wheats, einkorn, emmer and spelt, compared to modern wheat species, common and durum wheat. In fact, in our study, the emmer cultivars showed highest protein content (16.65 g kg^−^^1^, DM, on average) compared to modern durum wheat cultivars (15.14 g kg^−^^1^, DM, on average).

The yield-related parameters which resulted were significantly affected by species, genotype and ‘SxG’ and ‘SxGxY’ interactions whereas crop years have no effect on them. The higher values were observed in emmer wheat and in barley modern cultivars in both crop years, as a result of genetic gains in yield in both species [49,50,51,52]. In durum wheat, a different response was observed over the two crop years: Fortore showed the highest values of TW and TKW in 2016–2017 crop years and the lowest in the previous one, while the yield-related response of Cappelli was more stable in the two crop years. This confirms the behavior of old durum wheat cultivars which, although having a lower yield potential, are characterized by a lower sensitivity to environmental conditions and a greater stability of their productions [46,53].

### 3.2. Effects of Species, Genotype and Crop Year on the Content of Phenolic Compounds and Carotenoids, and Antioxidant Activities in Wholemeal

On wholemeal of all cultivars of the three species grown in two crop years, phenolics (TPC, TFC and TPAC) and TYPC, and DPPH- and ABTS-radical scavenging activities were determined and the effects of species (S), genotype (G), year (Y) and their combined interactions were measured by ANOVA (Table 3). S, G and ‘SxG’ were highly significant for all parameters (*p* < 0.001), whereas Y significantly affected only TPC, TPAC and TEAC and ‘SxGxY’ interactions were significant only for TPAC, TYPC, DPPH and TEAC.

Apart from the TYPC, barley was found to have significantly higher values for all the other parameters with respect to both wheats, of 38.5% on average for TPC and TFC, of 11% TPAC and of 24% on average for the two antioxidant capacities. The highest TPC, TFC and TPAC and antioxidant activities in barley are in agreement with previous results [36,54,55] confirming it to be an excellent dietary source of natural antioxidants with good health potential [56]. Although both ABTS and DPPH methods measured the antioxidant activity, the different levels can be explained by their different mechanisms [56,57]. The advantage of the ABTS radical is its high reactivity, and thus the likely ability to react with a broader range of antioxidants [58]. On the contrary, the DPPH method provides lower values related to Trolox than the ABTS method due to higher stability (and thus lower reactivity) of the DPPH radical. This agreed with durum and emmer wheat response, but not with barley. It is known that DPPH radical reacts with polyphenols (catechins, proanthocyanidins), but not with the phenolic acids and sugars [59]. That could explain the higher levels of DPPH compared to TEAC observed in barley, richest in polyphenols.

Durum wheat distinguished itself from the other species for the highest TYPC, with major levels in the modern cultivars as a result of breeding for this quality trait related to consumer preference for bright yellow color of pasta [19,20].

Among the cultivars, generally the highest values of all determinations were observed in the modern ones, with the only exception of the TPC for which the highest content was in the old wheat cultivars, Molisano and Cappelli [12,23]. A particular trend was observed for barley in which the modern cultivar Priora distinguished itself for the highest TPC values within and among species. Farther, Priora showed the highest values for all traits except for TYPC, higher in L94. This is consistent with what has been observed by other authors, as the L94, a black-colored cultivar, has more carotenoids compared to Priora, which has white seeds [18,40].

The crop years do not have a clear trend, and where the effect is significant, the response is variable, with higher values in the first year for TPAC and TEAC activity and in the second year for TPC. The lowest rainfall observed in the 2016/2017 crop season, in particular during the grain filling, from April to June (106.1 vs. 163.5 mm; Appendix A) could have determined a drought stress condition, resulting in a greater stress-induced synthesis of antioxidants, particularly polyphenols, to serve as free radical scavengers, mitigating oxidative and dehydration stress [60].

### 3.3. Effects of Species, Genotype and Crop Year on the Content of Some Phenolic Compounds and Carotenoids and Antioxidant Activities in Refined-Flours

TPC, TFC, TPAC and TYPC and DPPH- and ABTS-radical scavenging activities were analyzed on refined-flours of all genotypes of the three species grown in two crop years and the relative loss of each parameter compared to wholemeal was measured. The results are shown in Figure 1.

While the proanthocyanidins disappear completely in the refined flours of durum and emmer wheats, a consistent loss in their content was observed in barley, higher in the modern cultivar Priora than in the old L94 (73% vs. 56%). As observed by Irakli et al. [61], the flavanols were more concentrated in the bran, with a content three times higher than pearled flour. Furthermore, both barley cultivars showed the higher losses of TPC and TFC (55% and 57% on average, respectively), resulting in the highest reductions in DPPH and TEAC activities (63% and 52% on average, respectively), according to Van Hung [3], without consistent differences between old and modern cultivars. Conversely, the lowest decreases in TYPC were observed in this species (15%, on average) (Figure 1). Compared to barley, lower losses were generally observed in emmer and durum wheats. In these species, the responses in old and modern cultivars were different, except for DPPH activity and TPC in emmer (55% and 18%, on average, respectively). In durum wheat, the highest TPC and TFC losses were observed in the modern cultivar Fortore that, contrarily, showed the lower TYPC and both antioxidant activity losses. In wheat emmer, a different variation rate was observed for the other traits. In particular, the old cultivar Molisano showed the lowest decrease in TEAC scavenging activity compared to the modern PadrePio (15% vs. 39%, respectively). This agreed with Skendi et al. [62], who found in emmer landrace flours higher antioxidant activity than their commercial counterparts. As the TPC was lost to a minor extent in both wheats, their maintenance in refined-flour means it might be interesting to use this raw material to produce improved end-products. In particular, Cappelli was confirmed as a cultivar able to preserve useful compounds for health-promoting purposes [12,30].

### 3.4. Phenolic Acid Composition in Wholemeals and Refined-Flours

Soluble free and conjugated, and insoluble bound phenolic acids were investigated in wholemeals of the old and modern genotypes of the three species, and the results are shown in Table 4.

The ANOVA results were significant for all phenolics. Large variability was seen in the total content of these compounds across the species and the genotypes investigated. Overall, free and conjugated fractions resulted as more representative in emmer wheat (12%, on average) compared to durum wheat and barley (5%, on average), in line with the results of Andersson et al. [63], while the bound fraction was prevalent in barley and among wheats, in durum wheat, in according to Brandolini et al. [64]. The common phenolic acids were mainly ferulic acid, vanillin, coumaric acid and cis- and trans-cinnamic acids (Appendix A).

Insoluble bound phenolic acids represented 86.0%, on average, of the total phenolic acids, and ferulic acid was the most abundant in all species, with a variation range from 334.45 µg g^−^^1^ DW in barley to 293.80 µg g^−^^1^ DW in emmer, on average (Appendix A), according to previous studies [10,11,40,64]. Besides ferulic acid, sinapic acid was the second-most abundant phenolic acid, followed by coumaric acid and cis-cinnamic acid, and other minor components in common to all cultivars (i.e., vanillin, syringic acid, syringaldeide and vanillic acid) or genotype-dependent (protocatechuic acid, p-hydroxybenzoic acid, caffeic acid and trans-cinnamic acid). In barley and durum wheat, the highest-bound phenolic acids were observed in the old cultivars with the lowest levels in the second crop season (Table 4 and Appendix A). The highest values observed in the old cultivar Cappelli agreed with Menga et al. [12]. The opposite trend was observed for emmer wheat. Although phenolic acids have been involved in biotic and abiotic stress tolerance [1], the modern cultivars were mainly developed for yield performance and nutritional and qualitative traits, and not specifically for phenolic acid accumulation, and this could explain the major levels of these compounds in the old cultivars compared to modern ones [60]. As evidenced by other authors, a variability exists in the profile and quantity of phenolic acids among species. For instance, unlike Li et al. [11], who found sinapic acid in its free form only in durum wheat, in this study, sinapic acid was present in the free form only in emmer and in the bound form in all species, with prevalence in emmer wheat. In both fractions, the old cultivar Molisano had the highest content of this compound. The different phenolic acid profile of our results compared with those reported in the literature could be due to the different cultivars as well as the condition of extraction and the chromatography system.

The flavonoids, although mainly present in the stem and leaves of plants [65], were also found in bran and germ section of kernels [8]. Similarly to phenolic acids, the flavonoids are found in free, conjugated and bound form (Appendix A). In durum wheats, results showed they were absent. In free fraction, little quercetin amounts were observed in emmer Molisano in both crop years, while catechin was found in free and bound flavonoids in barley, with Priora having two-fold higher content compared to L94 in the free fraction. Naringenin is the common flavonoid to all fractions of emmer wheat and barley. Naringenin is a flavonoid belonging to flavanones subclass, widely spread in beans, citrus fruits, bergamot, tomatoes and other fruits, and in little amounts in cereals, with a possible role on plant growth, and stress responses in plants [66]. In barley, some authors [67,68] found naringenin, quercetin and catechin as potential biomarkers, involved in a significant in vitro reduction in the Fusarium graminearum, a devastating disease of Triticeae, causing yield losses, and also indirectly affecting the quality of grains.

A Pearson correlation was calculated among phenolic compounds and antioxidant activities (Appendix A), confirming that TPC, TFC, TPAC and catechin significantly contribute to both radical scavenging activities (*p* < 0.001). Other phenolic acids such as caffeic, syringic, coumaric and trans-cinnamic acids may perform a minor role in both antioxidant activities, with the exception of ferulic acid that positively correlated only with ABTS and sinapic acid that negatively affected the antioxidant potential. This agrees with Menga et al. [12] and Horvat et al. [69].

The refined flours have much lower phenolic acid content than the wholemeal. A total of 28.2% of total phenolic acids, on average, was found in refined flours in the three species (Appendix A) as also observed by Guan et al. [70]. The different losses compared to Menga et al. [12] could be explained by the inclusion in this study of other two species, besides durum wheat, showing different endogenous levels of phenolic acids.

Ferulic acid is the most abundant in all samples, accounting for up to 90% of total phenolic acids [62]. Our data are in according to Giordano et al. [71] and Skendi et al. [62], who observed an amount of free ferulic acid in refined-flours that does not surpass the value of 1.4 µg g^−^^1^ on average, whereas the amount of bound ferulic acid varied between 80.23 µg g^−^^1^ (durum wheat) and 128.94 µg g^−^^1^ (barley), suggesting that not only the kernel tissue but also the species and the genotype affect the amount of ferulic acid content.

In free fraction, other phenolic acids were identified as vanillin in durum wheat and barley, and syringic and p-coumaric acids in barley, with prevalence of old genotype. In the conjugated fraction, vanillin and p-coumaric acid are in common, while vanillic acid was found in durum wheat and barley and syringic acid in barley. The other two compounds in common for all species in insoluble bound phenolic acids were sinapic acid and p-coumaric acid. Interestingly, major phenolic acids were found in barley refined flour, particularly in L94, with the order ferulic acid > trans-cinnamic acid > sinapic acid > p-coumaric acid > caffeic acid > p-hydroxybenzoic acid > protocatechuic acid > vanillic acid. The highest levels of the insoluble bound form of phenolic acids were in the modern genotype for emmer wheat while; for the other species, results for the old were prevalent, reflecting the wholemeal trend. Similar to the phenolic acids, the flavonoids were found in free, conjugated and bound form. Among the studied species, only barley contained catechin in the bound flavonoid fraction, without differences between old and modern cultivars (Appendix A), in the range observed by Idehen et al. [24]. With wholemeal being higher in antioxidants, we will concentrate our attention on this type of flour becoming the raw material for biscuit making.

### 3.5. Principal Component Analysis of Grain and Wholemeal between Phenolics, Phenolic Acids, Quality and Yield-Related Traits

In order to analyze multiple variables in the grain and wholemeal of the three species and responses in two crop years, a principal component analysis (PCA) was undertaken, and results were reported by translating multiple data into a score plot and loading plot (Figure 2a,b). The principal component 1 (PC1) explained 44.4% of the total variance, whilst the principal component 2 (PC2) explained 18.5% of the variance (Appendix A). PC1 discriminated the species, barley genotypes being on the positive and durum wheat and emmer wheat on the negative axes. In turn, durum wheat and emmer wheat genotypes were separated along the PC2. Major antioxidant traits were mainly influenced by PC1, whereas TW, TKW and some other phenolics by PC2. In particular, the first factor was highly and positively associated with the TPC, TFC, TPAC, DPPH, TEAC, caffeic acid, syringic acid, p-coumaric acid, ferulic acid, trans-cinnamic acid, catechin and negatively with the TYPC, protocatechuic acid and sinapic acid. The second factor showed a positive association with yield-related traits (TW, TKW), and some phenolic acids (vanillic acid, vanillin, cis-cinnamic acid) and negatively with p-hydroxybenzoic acid and the flavonoid naringenin. Instead, both factors were negatively associated to PC.

The general inverse relation between the yield-related traits (TW and TKW) and p-hydroxybenzoic acid and naringenin observed along the Factor 2 indicated that yield-related traits are at odds with some phenolics, as previously observed in Menga et al. [12]. Regarding the cultivars, a discrimination was observed between the old and modern ones for emmer wheat along Factor 1. Instead, the crop years had a different trend only for durum wheat cultivars. On the basis of the PCA, the barley cultivars were closely related to most of the phytochemicals and to the antioxidant activities. These data are in agreement with previous studies [37,40,72]. For the emmer, the old cultivar Molisano is associated to naringenin and p-hydroxybenzoic acid, while the modern PadrePio by PC, protocatechuic acid and sinapic acid in the two years. Finally, durum wheat cultivars Fortore and Cappelli were distributed along the negative left quadrant and were associated with yield-related traits and with TYPC. It is noteworthy that TYPC in durum wheat is a criterion for the marketing and the nutritional quality of end-products, such as pasta [19,20].

### 3.6. Effect of Biscuit Processing in Phenolic Compound Level and Composition

For each species, a cultivar was selected for interesting levels of phytochemicals and used for biscuit making: in particular, Fortore (durum wheat) was high in TFC and TYPC, Molisano (emmer wheat) was high in TPC and free and conjugated flavonoids and L94 (barley) was high in conjugated and bounded flavonoids.

The experimental biscuits belonging to the three species were nutritionally superior to commercial control as they have highest TPC, TFC, TPAC and TYPC, as well as both antioxidant activities (Figure 3). Among the three species, durum wheat prevailed for the TYPC, while barley was confirmed to have the best performance for all the other analyzed parameters.

With respect to the corresponding wholemeal, flavonoids decreased in all samples, while in Fortore and Molisano an increase in total polyphenols and proanthocyanidins was observed (24% and 55% on average, respectively). A greater increase was found for these traits in the CTRL. On the contrary, they decreased in L94 (18% and 31%, respectively) (Appendix A; Figure 3). The antioxidant activities reflect this trend, with results partially overlapped with the results of Li et al. [73] on muffins. In all biscuits, a positive effect of baking on the free, conjugated and bound phenolic acids was observed, confirming the general response reported by Abdel-Aal and Rabalski [21] for cookies and muffins.

As product-making processes were the same for all our biscuits, the unique difference in the recipe was the flour belonging to the different cultivars that may have contributed to changes in phenolic contents among the end-products.

Ferulic acid was the principal phenolic acid in the free, conjugated or bound extracts of the end-products, showing the highest values in bound form (about thirty-fold higher when compared to control biscuit). With respect to corresponding wholemeal, an increase in free ferulic acid content was observed in Fortore and L94 (41% and 56%, respectively), the conjugated form was negatively affected, particularly in barley and emmer wheat (45% of loss, on average) while bound fraction decreased in the same extent in all samples (47% on average) (Table 5 and Appendix A). The increment observed in free ferulic acid could be due to the release of bound forms from the food matrix during the baking process [21,43].

Ferulic acid, vanillic acid and vanillin were the common phenolic acids in all extracts of the three products. Vanillin being a component of the biscuit recipe, its high levels are not dependent on baking conditions or the flour, and they have no relevance. Major phenolic acids were found in the bound fraction (p-coumaric, sinapic, p-hydroxybenzoic, syringic, caffeic acids and syringaldeide), while p-coumaric and sinapic acids were found in the conjugated fraction. In the commercial control, only p-coumaric acid in the bound fraction was detected.

In general, other than the release of bound phenolics from the food matrix, different mechanisms could be involved in changing phenolic acids during baking, such as polymerization and oxidation of phenolics, thermal degradation and production of Maillard reaction products, as supposed by other authors [74,75].

### 3.7. Sensory Biscuits Profile

The degree of liking of the biscuits obtained from different species in comparison to commercial control was assessed by the consumer test evaluation based on its sensory appeal (color, odor, sweetness, crumbliness, crispness, flavor and overall acceptability). A radar graph represents the sensory data (Figure 4). Sweetness, crumbliness, flavor and overall acceptability in Molisano were rated by the consumers with even higher scores than the control sample. Then, Fortore biscuits emerged for crispness and odor and finally, L94 was not appreciated, except for crumbliness. In all examined samples, the crust color was from bright yellow in biscuit control to brownish yellow. Color and external aspect of biscuits could be affected by reducing sugars, which caramelize during the baking process producing brown color [76]. Considering sweetness, the opposite trend was evidenced for Molisano and L94 showing the highest and lowest values, respectively. This diverse response could be due to different sugar content and composition of the flours influencing other than the sensory characteristics, also the structure and texture of dough and subsequent cooking performance [77].

The aromatic attributes scores were quite different referred to odor (smell) and flavor (taste). The odor was better in control biscuits while the L94 was the worst. Instead, for flavor considered the whole package as combination of taste, odor and chemical sensations, the emmer, the durum and the control biscuit samples clearly differed from the barley one. Although barley was one of the best sources of phytochemicals, it was not appreciated by consumers and not marketable.

## 4. Conclusions

This work provides new insights into the quali-quantitative composition of some cereal bioactive metabolites in relation to their potential antioxidant activity in wholemeal and cereal-based products. Wholemeal of cereals represents a rich source of phenolics, mostly phenolic acids and flavonoids. Among the studied species, the barley resulted to show the best performance for all traits, except for total carotenoids. The phenolic acids were more representative in emmer in free and conjugated fractions, while the bound fraction, representing the 86% of the total, was prevalent in barley and durum wheat. Considering the old and the modern cultivars, generally the modern ones contained higher levels of antioxidants, except for TPC and TPAC that, instead, were more affected by growing season.

As the baking processes resulted in a loss of phenolics in biscuits as compared to wholemeal flour, the choice of raw materials richest in these compounds becomes crucial in obtaining better final products. In this optic, although barley biscuits were richer in antioxidants, the less consumer acceptance could limit its diffusion. Considering the popularity of this product, the optimization of ingredients or the blending of wheat flour with selected fractions of barley to obtain enriched biscuits should always go hand-in-hand with sensory evaluation, to reach health benefits and to be easily marketable.

## Figures and Tables

**Figure 1 foods-12-02551-f001:**
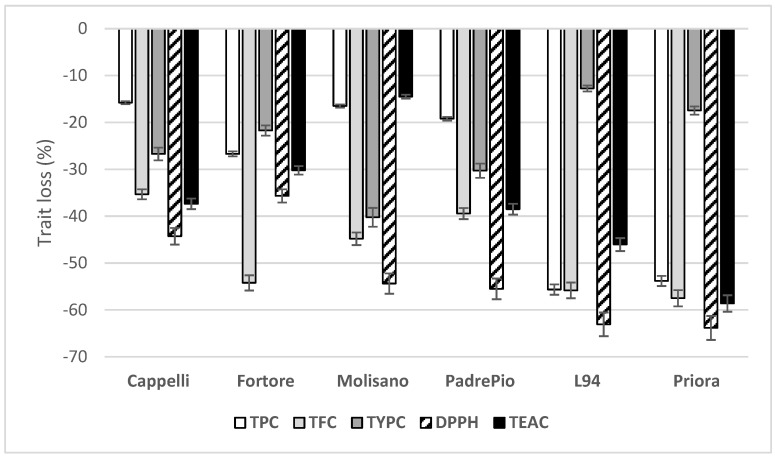
Variation in total polyphenols (TPC), flavonoids (TFC), carotenoids (TYPC) and antioxidant activities (DPPH and TEAC) in refined flours, for all cultivars of the three species analyzed in two crop years. Durum wheats: Cappelli and Fortore; emmer wheat: Molisano and PadrePio; barley: L94 and Priora.

**Figure 2 foods-12-02551-f002:**
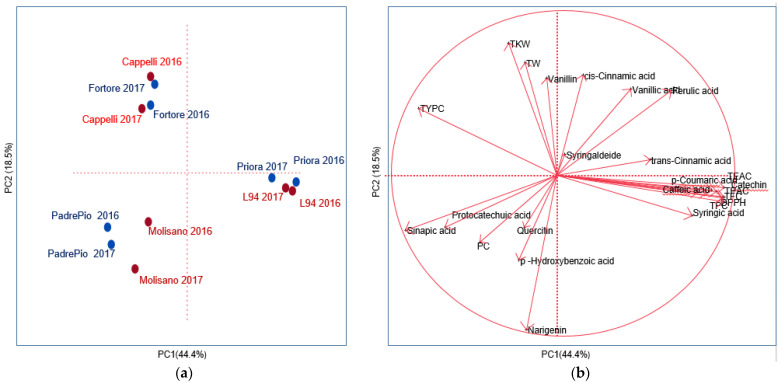
Principal component analysis (PCA) score plot (**a**) and loading plot (**b**) of the trait analyzed in the three species. In red, the old, and in blue, the modern cultivars belonging to the three species. TPC = total polyphenols; TFC = total flavonoids; TPAC = total proanthocyanidins; TYPC = total carotenoids; DPPH = antioxidant activity; TEAC = antioxidant activity; PC = protein content; TW = test weight; TWK = thousand kernel weight.

**Figure 3 foods-12-02551-f003:**
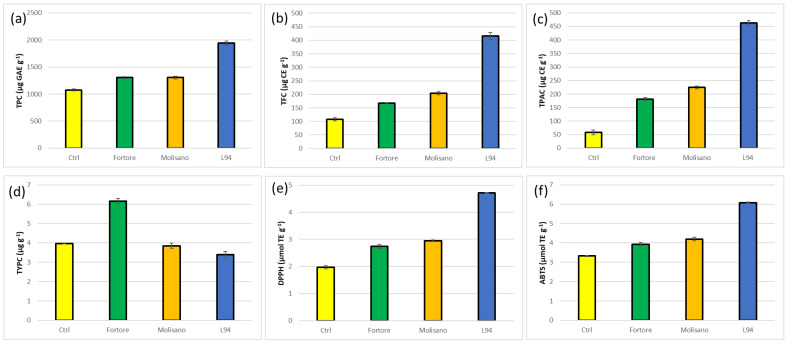
Total polyphenols (TPC) (**a**), total flavonoids (TFC) (**b**), total proanthocyanidins (TPAC) (**c**), total carotenoids (TYPC) (**d**), antioxidant activity (DPPH) (**e**), antioxidant activity (ABTS) (**f**) of biscuits from commercial CTRL (yellow), Fortore (Durum wheat, green), Molisano (Emmer wheat, orange) and L94 (Barley, blue). (Means and standard deviations; values expressed on dry matter).

**Figure 4 foods-12-02551-f004:**
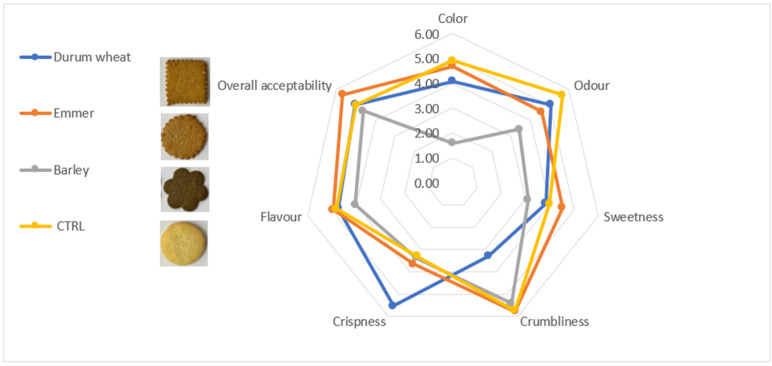
Radar plot obtained from consumer test evaluation of different biscuits.

**Table 1 foods-12-02551-t001:** Area of origin and pedigree of the genotypes used in this study.

Taxonomic Classification	Accession	Cultivar/Landrace-Origin	Year of Release	Genotype
**Durum wheat**	Cappelli	Cultivar-Selection from Tunisian population ‘Jean Retifah’	1915	Old	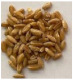
Fortore	Cultivar–Capeiti-8/Valforte	1995	Modern	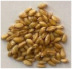
**Emmer wheat**	Molisano	Landrace-Molise region, Central Italy	//	Old	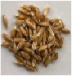
PadrePio	Cultivar–Simeto/Molise	2016	Modern	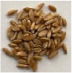
**Barley**	L94	Ethiopian landrace line (black and naked grains)	//	Old	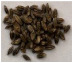
Priora	Cultivar–Arda/Mondo (white and naked grains)	2000	Modern	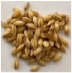

**Table 2 foods-12-02551-t002:** Qualitative and yield-related traits in wholegrain samples.

Species	Genotype	Type	Year	PC (g kg^−1^, DM)	TW (Kg hL^−1^)	TKW (g)
**Interaction of Species x Genotype x Year effects (SxGxY)**						
*Durum wheat*	Cappelli	Old	2015/16	15.90	81.43 a,b	54.60 a,b
*Durum wheat*	Cappelli	Old	2016/17	16.50	81.18 a,b	52.43 b,c
*Durum wheat*	Fortore	Modern	2015/16	14.30	78.16 b,c	48.33 c,d
*Durum wheat*	Fortore	Modern	2016/17	13.83	82.51 a	57.73 a
*Emmer wheat*	Molisano	Old	2015/16	14.83	73.06 e,f	42.30 e,f
*Emmer wheat*	Molisano	Old	2016/17	16.70	69.37 f	40.13 f
*Emmer wheat*	PadrePio	Modern	2015/16	16.63	80.98 a,b	45.60 d,e
*Emmer wheat*	PadrePio	Modern	2016/17	18.40	77.96 b,c	46.87 d
*Barley*	L94	Old	2015/16	13.87	74.60 c–e	40.67 f
*Barley*	L94	Old	2016/17	14.87	75.21 c–e	40.07 f
*Barley*	Priora	Modern	2015/16	14.93	80.15 a,b	47.07 d
*Barley*	Priora	Modern	2016/17	16.23	78.93 a–c	49.03 c,d
	*F_(2.24)_*			*0.99*	*4.0*	*8.56*
	*p* value			n.s.	***	****
**Interaction of Species x Genotype effects (SxG)**						
*Durum wheat*	Cappelli	Old		16.20 a,b	81.31 a	53.52 a
*Durum wheat*	Fortore	Modern		14.07 d	80.34 a	53.03 a
*Emmer wheat*	Molisano	Old		15.77 b,c	71.21 c	41.22 c
*Emmer wheat*	PadrePio	Modern		17.52 a	79.47 a	46.23 b
*Barley*	L94	Old		14.37 c,d	74.90 b	40.37 c
*Barley*	Priora	Modern		15.58 b–d	79.54 a	48.05 b
	*F_(2.30)_*			*16.67*	*17.24*	*7.53*
	*p* value			*****	*****	****
**Single effect (Species) (S)**						
	*Durum wheat*			15.13 b	80.82 a	53.28 a
	*Emmer wheat*			16.64 a	75.34 b	43.73 b
	*Barley*			14.98 b	77.22 b	44.21 b
	*F_(2.33)_*			*6.56*	*7.98*	*24.89*
	*p* value			**	***	***
**Single effect (Genotype) (G)**						
	Old genotypes			15.44	75.81 b	45.03 b
	Modern genotypes			15.72	79.78 a	49.11 a
	*F_(1.34)_*			*0.33*	*11.28*	*5.05*
	*p* value			n.s.	**	*
**Single effect (Year) (Y)**						
	2015/16			15.08 b	78.06	46.43
	2016/17			16.09 a	77.53	47.71
	*F_(1.34)_*			*5.03*	*0.15*	*0.44*
	*p* value			*	n.s.	n.s.

PC = protein content; TW = test weight; TKW = thousand kernel weight. For each parameter, different letters indicate significant differences according to the Tukey’s test (*p* ≤ 0.05). *, **, ***, significant at 0.05, 0.01 and 0.001 probability level, respectively; n.s., not significant.

**Table 3 foods-12-02551-t003:** Mean values of Total polyphenols (TPC), flavonoids (TFC), proanthocyanidins (TPAC), carotenoids (TYPC) and antioxidant activities (DPPH and TEAC), for wholemeal of all genotypes of the three species analyzed in two crop years.

Species	Genotype	Type	Year	TPC (µg GAE g^−1^)	TFC (µg CE g^−1^)	TPAC (µg CE g^−1^)	TYPC (µg g^−1^)	DPPH (µmol TE g^−1^)	TEAC (µmol TE g^−1^)
**Interaction of Species x Genotype x Year effects (SxGxY)**									
*Durum wheat*	Cappelli	Old	2015/16	1065.09	288.82	118.52 d	5.66 c	1.98 c,d	2.52 e–g
*Durum wheat*	Cappelli	Old	2016/17	1173.80	276.07	115.25 d	5.75 c	2.09 c,d	2.77 d
*Durum wheat*	Fortore	Modern	2015/16	896.52	313.82	145.16 d	7.58 a	1.74 d	2.71 d,e
*Durum wheat*	Fortore	Modern	2016/17	925.88	301.71	128.32 d	6.79 b	1.81 c,d	2.57 e,f
*Emmer wheat*	Molisano	Old	2015/16	1067.93	289.47	113.09 d	4.49 e	1.99 c,d	2.34 f,g
*Emmer wheat*	Molisano	Old	2016/17	1167.83	314.84	125.53 d	4.99 d	2.20 c,d	2.28 g
*Emmer wheat*	PadrePio	Modern	2015/16	1037.73	295.60	132.68 d	5.71 c	2.16 c,d	2.31 g
*Emmer wheat*	PadrePio	Modern	2016/17	1066.12	325.12	134.86 d	6.08 c	2.27 c	2.37 f,g
*Barley*	L94	Old	2015/16	2565.04	717.66	824.41 c	3.78 f	8.85 b	8.10 c
*Barley*	L94	Old	2016/17	2668.70	731.91	822.78 c	3.87 f	8.38 b	8.16 c
*Barley*	Priora	Modern	2015/16	2704.27	871.41	1639.18 a	2.92 g	11.43 a	10.95 a
*Barley*	Priora	Modern	2016/17	2823.60	863.26	1314.95 b	3.10 g	11.06 a	10.06 b
	*F_(2.24)_*			*1.79*	*1.2*	*73.39*	*7.32*	*0.23*	*10.89*
	*p* value			n.s.	n.s.	*****	*****	****	*****
**Interaction of Species x Genotype (SxG)**									n.s.
*Durum wheat*	Cappelli	Old		1119.44 c	282.45 d	116.88 c	5.70 b	2.04 c,d	2.64 c
*Durum wheat*	Fortore	Modern		911.20 e	307.76 c	136.74 c	7.19 a	1.78 d	2.64 c
*Emmer wheat*	Molisano	Old		1117.88 c	302.15 c	119.31 c	4.74 c	2.09 c,d	2.31 d
*Emmer wheat*	PadrePio	Modern		1051.93 d	310.36 c	133.77 c	5.89 b	2.22 c	2.34 d
*Barley*	L94	Old		2616.87 b	724.78 b	823.59 b	3.83 d	8.62 b	8.13 b
*Barley*	Priora	Modern		2763.93 a	867.34 a	1477.06 a	3.01 e	11.24 a	10.51 a
	*F_(2.30)_*			*82.11*	*122.99*	*1231.45*	*176.87*	*210.67*	*283.97*
	*p* value			*****	*****	*****	*****	*****	*****
**Single effect (Species) (S)**									
	*Durum wheat*			1015.32 c	295.10 c	126.81 b	6.45 a	1.91 c	2.46 b
	*Emmer wheat*			1084.90 b	306.26 b	126.54 b	5.32 b	2.16 b	2.33 b
	*Barley*			2690.40 a	796.06 a	1150.33 a	3.42 c	9.93 a	9.32 a
	*F_(2.33)_*			*9224.26*	*7527.75*	*12,747.46*	*1068.10*	*7155.16*	*9508.79*
	*p* value			***	***	***	***	***	***
**Single effect (Genotype) (G)**									
	Old genotypes			1618.06 a	436.46 b	353.26 b	4.76 b	4.25 b	4.36 b
	Modern genotypes			1575.69 b	495.15 a	582.52 a	5.36 a	5.08 a	5.16 a
	*F_(1.34)_*			*13.83*	*237.67*	*1438.67*	*126.27*	*177.7*	*294.06*
	*p* value			***	***	***	***	***	***
**Single effect (Year) (Y)**									
	2015/16			1556.10 b	462.80	495.50 a	5.02	6.49	4.82 a
	2016/17			1637.66 a	468.82	440.28 b	5.10	4.64	4.70 b
	*F_(1.34)_*			*51.24*	*2.5*	*83.47*	*1.90*	*0.81*	*6.55*
	*p* value			*****	n.s.	*****	n.s.	n.s.	***

For each parameter, different letters indicate significant differences according to the Tukey’s test (*p* ≤ 0.05). *, **, ***, significant at 0.05, 0.01 and 0.001 probability level, respectively; n.s., not significant.

**Table 4 foods-12-02551-t004:** Mean value of total content of soluble free and conjugated, and insoluble bound phenolic acids and flavonoids in the wholemeal of old and modern genotypes of the three species analyzed in two crop years. Data are expressed as µg g^−1^, DM.

Species	Genotype	Crop Years	TSF Phenolic Acids	TSF Flavonoids	TSC Phenolic Acids	TSC Flavonoids	TIB Phenolic Acids	TIB Flavonoids
Durum wheat	**Cappelli**	2015/16	20.11 c,d	n.d.	26.83 de	n.d.	465.87 a	n.d.
		2016/17	20.58 c,d	n.d.	24.72 e	n.d.	389.53 b	n.d.
	**Fortore**	2015/16	13.01 g	n.d.	29.06 d,e	2.19 a	384.25 b	n.d.
		2016/17	14.12 f,g	n.d.	38.11 b	1.71 b	314.66 b	n.d.
Emmer wheat	**Molisano**	2015/16	23.17 c	5.37 d	36.18 b,c	0.44 d	238.06 c	n.d.
		2016/17	19.11 c–f	5.57 d	31.44 d	0.59 c	266.97 c	n.d.
	**PadrePio**	2015/16	72.73 a	2.47 e	52.24 a	n.d.	242.67 c	0.74 c,d
		2016/17	43.32 b	2.37 e	42.07 b	n.d.	276.30 c	1.12 c
Barley	**L94**	2015/16	19.20 c–e	18.90 c	27.26 d,e	0.55 c,d	502.04 a	5.19 a,b
		2016/17	43.03 b	25.80 b	30.06 d,e	0.51 c,d	423.87 a,b	4.49 b
	**Priora**	2015/16	14.70 e–g	52.76 a	27.72 d,e	n.d.	482.55 a	5.17 a,b
		2016/17	15.78 d–g	53.89 a	25.45 e	n.d.	375.15 b	5.65 a
*F_(2.24)_ (SxGxY)*			*48.96*	*11.96*	*14.85*	*4.49*	*1.00*	*3.84*
*p* value			*****	*****	*****	***	***	***

TSF = Total Soluble Free; TSC = Total Soluble Conjugated; TIB = Total Insoluble Bound. For each parameter, different letters indicate significant differences according to the Tukey’s test (*p* ≤ 0.05). * and *** significant at 0.05 and 0.001 probability level, respectively.

**Table 5 foods-12-02551-t005:** Comparison of phenolic acid composition, free- and conjugated-soluble and insoluble bound, in biscuits derived from the best genotype choosen within each of the three species and from commercial control. (Means and standard deviations; values expressed as µg g^−1^, DM).

	Durum Wheat			Emmer Wheat			Barley			Commercial		
	Fortore			Molisano			L94			Control		
	Soluble Free	Soluble Conjugated	Insoluble Bound	Soluble Free	Soluble Conjugated	Insoluble Bound	Soluble Free	Soluble Conjugated	Insoluble Bound	Soluble Free	Soluble Conjugated	Insoluble Bound
Vanillic acid	38.24 ± 0.94	37.87 ± 1.43	9.75 ± 0.47	12.96 ± 0.01	19.17 ± 1.24	5.16 ± 0.43	15.20 ± 1.02	19.17 ± 1.24	6.4 ± 0.49	4.20 ± 0.4	52.91 ± 0.15	1.23 ± 0.11
Vanillin	1127.44 ± 8.49	661.12 ± 21.87	40.13 ± 1.06	1237.92 ± 2.26	876.22 ± 28.42	29.89 ± 1.77	1149.56 ± 5.83	871.23 ± 35.49	29.61 ± 1.19	1400.84 ± 12.16	868.12 ± 29.77	38.59 ± 0.41
Ferulic acid	6.52 ± 0.17	11.23 ± 0.49	327.04 ± 22.02	5.36 ± 0.57	9.17 ± 0.38	265.41 ± 5.2	5.72 ± 0.41	9.20 ± 0.34	347.09 ± 0.07	1.12 ± 0.11	0.88 ± 0.04	10.91 ± 1.13
Sinapic acid	1.40 ± 0.06	5.84 ± 0.34	10.03 ± 0.49	4.08 ± 0.82	5.47 ± 0.64	9.25 ± 0.15	n.d.	12.16 ± 0.30	11.29 ± 0.09	n.d.	n.d.	n.d.
Catechin	n.d.	n.d.	n.d.	n.d.	n.d.	n.d.	n.d.	n.d.	n.d.	n.d.	n.d.	n.d.
p-Hydroxybenzoic acid	n.d.	n.d.	1 ± 0.05	n.d.	n.d.	0.91 ± 0.03	n.d.	n.d.	0.75 ± 0.07	n.d.	n.d.	n.d.
Syringic acid	n.d.	n.d.	1.04 ± 0.15	n.d.	n.d.	0.8 ± 0.07	n.d.	n.d.	1.16 ± 0.39	n.d.	n.d.	n.d.
p-Coumaric acid	n.d.	1.39 ± 0.01	9.36 ± 0.72	n.d.	0.83 ± 0.04	16.61 ± 0.19	n.d.	8.27 ± 0.15	12.49 ± 1.22	n.d.	n.d.	9.27 ± 0.04
Syringaldeide	n.d.	n.d.	2.89 ± 0.32	n.d.	n.d.	3.59 ± 0.05	n.d.	n.d.	3.87 ± 0.07	n.d.	n.d.	n.d.
Caffeic acid	n.d.	n.d.	0.87 ± 0.13	n.d.	n.d.	1.87 ± 0.07	n.d.	n.d.	0.87 ± 0.13	n.d.	n.d.	n.d.

## Data Availability

The data presented in this study are available on request from the corresponding author. The data are not publicly available.

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
