# Peer review of "Antioxidants and Phenolic Acid Composition of Wholemeal and Refined-Flour, and Related Biscuits in Old and Modern Cultivars Belonging to Three Cereal Species"

_foods, 2023, doi:10.3390/foods12132551_

Round 1

Reviewer 1 Report

PLEASE LOOK at the attachment

Author Response

We thank the Reviewer for the considerations. The manuscript has been improved following his/her suggestions, with the indications of our modifications also given in our replies. The paper has been revised to improve the English language.

- Lack of details when it comes to equipment (type, model, city, country) I am asking the Authors to review the methodology once again

Our reply: We have checked all the equipment in the Materials and Methods. The information resulted correctly reported (i.e., line 136 for NIR, line 143 for Cyclotec, lines 145-146 Bona milling, lines 153-154 for stone mill, line 160 for electric mixer, line 163 for oven, line 190 for HPLC system and lines 192-193 for C18 column). For the other methodologies, the authors refer to the methods reported in the relative references.

- Please perform FRAP because this designation is complementary to ABTS and DPPH. DPPH and ABTS indicate only one possibility of polyphenols acting as antioxidants. In nutritional matrices (raw  materials and product) antioxidants act in several directions, to talk about their activity, at least two of them should be analyzed. That's why I propose FRAP ( Fe reduction catalyzing Fenton reactions).

Our reply: We agree with the Reviewer’s suggestion to evaluate another method to explore antioxidant capacity but, due to the tight time frame for the review, we are unable to respond to this request. Anyway, our choice has been supported by literature reviewed in Munteanu and Apetrei, 2021 (https://doi.org/10.3390/ijms22073380) stating that “while the ABTS and CUPRAC tests can measure both hydrophilic, and lipophilic antioxidants, some methods only measure hydrophilic antioxidants (FRAP and Folin−Ciocalteu), and others only apply to hydrophobic systems (DPPH)”.

We will keep your suggestion in mind for the future.

- I am asking the Authors to discuss in detail how much the percentage of a given ingredient changes in samples. The following sentence is an example. „Apart from the TYPC, barley was found to have significantly higher values ( HOW MUCH PERCENT) for all the other parameters, followed by emmer wheat”.

Our reply: We have revised the manuscript providing information on the percentage derived by analyzing different cultivars for the traits studied, modifying the text as: “Apart from the TYPC, barley was found to have significantly higher values for all the other parameters respect to both wheats, of 38.5% on average for TPC and TFC, of 11% TPAC and of 24% on average for the two antioxidant capacities”. (Now lines 286-287).

Other details have already been filled in the text by making the improvements proposed by Reviewers 2 and 3.

- Conclusions should have significant data

Our reply: We have added some information in the text. (Now lines 577-579).

Reviewer 2 Report

1.      In tables, authors represent data in species or genotypes, while in the discussion section, authors are using barley, emmer, cappelli etc. The authors should follow a single pattern in the manuscript, either using genotypes or crop names.

2.       Include the statistical significance in protein content in Table 2 between the old and new genotypes.

3.       In Table 2, Include the year in section 2 (Interaction of Species x Genotype effects (SxG)).

4.      In lines 233-234, the authors mentioned (Among durum wheat, the old cultivar was characterized by a higher protein percentage than the modern one, because of its lower grain yield). How the yield affects the protein content? Discuss in detail.

5.      In Table 3, Include the statistical significance in TPC and TFC between genotypes.

6.      The table 2 and 3 show single effect genotype and year, but genotype and species are not clear. The authors should include the genotype and species.

7.       In section 3.2, (line no 265-291), the authors have discussed about the variation in the TPC, TFC, TPAC etc, between the different species but the discussion between old and modern is lacking, similarly, in sections 3.3 and 3.4. Further, the authors must include the percentage increase or decrease in the TPC, TFC, TPAC between or within the species in the discussion section.

8.      In Figure 1 include the control (wholemeal), year is not included, and present the graph control vs different species.

9.       Provide the chromatogram for the phenolic and flavonoid compounds in the supplementary data.

10.  In figure 3, includes the units on Y-axis in the figure.

11.  The discussion in section 3.6 is not supported by data.

12.  The discussion section in the manuscript can be improved; the authors should enrich the section with literature.

Author Response

We thank the Reviewer for the exhaustive overview of the manuscript. The manuscript has been improved following his/her suggestions, with the indications of our modifications also given in our replies.

-In tables, authors represent data in species or genotypes, while in the discussion section, authors are using barley, emmer, cappelli etc. The authors should follow a single pattern in the manuscript, either using genotypes or crop names.

Our reply: We have checked all the Tables standardizing the name of species with those reported in Results and Discussion section.

-Include the statistical significance in protein content in Table 2 between the old and new genotypes.

Our reply: We thank the Reviewer for the observation. The differences between old and modern genotypes resulted not significant (n.s.) for protein content (Table 2). 

-In Table 2, Include the year in section 2 (Interaction of Species x Genotype effects (SxG)).

Our reply: In Table 2, the effect of year has been included in the first section (Interaction of Species x Genotype x Year effects). As described in Subsection 2.6 of the manuscript, one-way analysis of variance (ANOVA) was performed to estimate differences ascribable to the effects of single factor such as species (S), genotype (G), or year (Y), while two-factor ANOVA was applied to study the effect of the ‘SxG’ and ‘SxGxY’ interactions.

-In lines 233-234, the authors mentioned (Among durum wheat, the old cultivar was characterized by a higher protein percentage than the modern one, because of its lower grain yield). How the yield affects the protein content? Discuss in detail.

Our reply: Many thanks for Reviewer’ suggestions. We have improved the text as follows “Among durum wheats, the old cultivar was characterized by higher protein percentage compared with modern one, as a consequence of breeding programs for higher yields at the expense of grain quality [31,45,46]. The yield increase, essentially due to a greater carbon availability to the grains, is accompanied by the decrease of protein content, by dilution effects [47]”. (Now lines 249-253).

Furthermore, we have added a new reference “Lawlor, D.W., 2002. Carbon and nitrogen assimilation in relation to yield: mechanisms are the key to understanding production systems. J. Exp. Bot. 53, 773–787”.

-In Table 3, Include the statistical significance in TPC and TFC between genotypes.

Our reply: The statistical significance in TPC and TFC between old and modern genotypes was just explained by p value and Tukey’s test in the box n. 4.  

-The table 2 and 3 show single effect genotype and year, but genotype and species are not clear. The authors should include the genotype and species.

Our reply: In both Tables 2 and 3, the single effect of species (durum wheat, emmer wheat and barley) (box n. 3), genotype (old and modern) (box n. 4) and year (box n. 5) as well as the effect of the interaction of species x genotype (box n. 2) have been shown.

-In section 3.2, (line no 265-291), the authors have discussed about the variation in the TPC, TFC, TPAC etc, between the different species but the discussion between old and modern is lacking, similarly, in sections 3.3 and 3.4. Further, the authors must include the percentage increase or decrease in the TPC, TFC, TPAC between or within the species in the discussion section.

Our reply: We thank the reviewer for his/her observations. We want to underline that in refined flours respect to wholemeal, a reduction of the levels of all parameters was always found, as specified in line 323 and Y-axis of Figure 1. The percentage of losses were also added in Sections 3.3 and 3.4.

Regarding Section 3.2 we have re-visited the previous text reported in lines 295-300 as “Among the cultivars, generally the highest values of all determinations were observed in the modern ones, with the only exception of the TPC for which the highest content was in the old wheat cultivars, Molisano and Cappelli [12,23]. A particular trend was observed for barley in which the modern cultivar Priora distinguished itself for the highest TPC values within and among species. Farther, Priora showed the highest values for all traits except for TYPC, higher in L94. This is consistent with what has been observed by other authors, as the L94, that is a black coloured cultivar, has more carotenoids respect to Priora, that has white seeds [18,40]. (Now lines 303-310).

Section 3.3 was also revised and implemented. So, the lines 319-333 were substituted as “While the proanthocyanidins disappear completely in the refined-flours of durum and emmer wheats, a consistent loss in their content was observed in barley, higher in the modern cultivar Priora than in the old L94 (73% vs. 56%). As observed by Irakli et al. [60], the flavanols were more concentrated in the bran, with a content three times higher than pearled flour. Furthermore, both barley cultivars showed the higher losses of TPC and TFC (55% and 57% on average, respectively), resulting in the highest reductions of DPPH and TEAC activities (63% and 52% on average, respectively), according to Van Hung [3], without consistent differences between old and modern cultivars. Conversely, the lowest decreases in TYPC were observed in this species (15%, on average) (Figure 1). Respect to barley, lower losses were generally observed in emmer and durum wheats. In these species, the response in old and modern cultivars were different, except for DPPH activity and TPC in emmer (55% and 18%, on average, respectively). In durum wheat, highest TPC and TFC losses were observed in modern cultivar Fortore that, contrarly, showed the lower TYPC and both antioxidant activity losses. In wheat emmer, different rate were observed for the other traits. In particular, the old cultivar Molisano showed the lowest decrease of TEAC scavenging activity respect to the modern PadrePio (15% vs. 39%, respectively). This agreed with Skendi et al. [61] who found in emmer landrace flours higher antioxidant activity than their commerical counterparts. As the TPC was lost in a minor extent in both wheats, their maintenance in refined-flours might be interesting to use this raw material to produce improved end-products. In particular, Cappelli was confirmed as a cultivar able to preserve useful compounds for health-promoting purposes [12,30]. (Now lines 330-352).

Also Section 3.4 was revisioned. (Now lines 389-393 and 397-398). A new citation has been added, Harvat et al., 2020.

-In Figure 1 include the control (wholemeal), year is not included, and present the graph control vs different species.

Our reply: As year is not significant for major traits reported in Figure 1, the data were simplified and reported as mean values of the two years. The data were just presented as variation (%) for each trait in refined-flour respect to relative wholemeal (ctrl). The former data have been already shown in Table 3.

-Provide the chromatogram for the phenolic and flavonoid compounds in the supplementary data.

Our reply: We have inserted in Figure S1 an example of phenolic and flavonoid chromatograms during the whole cereal food supply chain, also implementing section 2.4.3. (Lines 204-205).

-In figure 3, includes the units on Y-axis in the figure.

Our reply: Done.

-The discussion in section 3.6 is not supported by data.

Our reply: Now the section has been supported by data. In adding, two mistakes have been found in Figure 3a and in Table 5 and they have been corrected.

-The discussion section in the manuscript can be improved; the authors should enrich the section with literature.

Our reply: During the revision, in the Results and Discussion Section we have implemented the discussion by inserting, where necessary, other citations or mention again other just reported references.

Reviewer 3 Report

After carefully reading the manuscript entitled: "Antioxidants and phenolic acid composition of wholemeal and refined-flour, and related biscuits in old and modern cultivars belonging to three cereal species" it can be concluded that the authors spent a lot of time and effort in conducting experiments and writing an article. The topic is interesting and novel. The paper is nicely written and of good quality. However, a few things could be improved. Below are remarks and suggestions.

  1. The pictures in Table 1 should be given individually in a separate file since nothing can be seen this way.
  2. Figure 3 is in low resolution and is not clear. Reconsider improving it and/or adding to the supplementary document in full size.
  3. What would complete this work is to make a correlation between phenolic compounds and antioxidant activities. Depending on the obtained results (if possible), consider introducing another method for antioxidant potential.

Author Response

We would like to thank the reviewer for all considerations.

-The pictures in Table 1 should be given individually in a separate file since nothing can be seen this way.

Our reply: Done.

-Figure 3 is in low resolution and is not clear. Reconsider improving it and/or adding to the supplementary document in full size.

Our reply: We have improved the Figure 3, considering all suggestions of Reviewers 2 and 3. Furthermore, there are no supplementary document related to this Figure.

-What would complete this work is to make a correlation between phenolic compounds and antioxidant activities. Depending on the obtained results (if possible), consider introducing another method for antioxidant potential.

Our reply: We thank the Reviewer for his/her suggestions. We have reported the correlations of phenolic compounds with the antioxidant activities as Suppl. Material (Table S3). The related comment has been added in the paragraph 3.4 (now lines 415-421), inserting a new citation (Horvat et al., 2020) and the statistical section in Materials and Methods has been implemented.

The antioxidant activities applied in this manuscript are widely used in cereals. As answered to Reviewer 1, our choice has been supported by literature reviewed in Munteanu and Apetrei, 2021 (https://doi.org/10.3390/ijms22073380) stating that “while the ABTS and CUPRAC tests can measure both hydrophilic, and lipophilic antioxidants, some methods only measure hydrophilic antioxidants (FRAP and Folin−Ciocalteu), and others only apply to hydrophobic systems (DPPH)”. We will consider this suggestion in the future.

Round 2

Reviewer 1 Report

I do not  have  any remarks